# Single-chain tandem macrocyclic peptides as a scaffold for growth factor and cytokine mimetics

Kenichiro Ito [1✉], Yoshihiko Matsuda[1], Ayako Mine[1], Natsuki Shikida[1], Kazutoshi Takahashi [1], Kyohei Miyairi[1], Kazutaka Shimbo[1], Yoshimi Kikuchi[1] & Atsushi Konishi[1]

Mimetics of growth factors and cytokines are promising tools for culturing large numbers of cells and manufacturing regenerative medicine products. In this study, we report single-chain tandem macrocyclic peptides (STaMPtides) as mimetics in a new multivalent peptide format. STaMPtides, which contain two or more macrocyclic peptides with a disulfide-closed back-bone and peptide linkers, are successfully secreted into the supernatant by *Corynebacterium glutamicum*-based secretion technology. Without post-secretion modification steps, such as macrocyclization or enzymatic treatment, bacterially secreted STaMPtides form disulfide bonds, as designed; are biologically active; and show agonistic activities against respective target receptors. We also demonstrate, by cell-based assays, the potential of STaMPtides, which mimic growth factors and cytokines, in cell culture. The STaMPtide technology can be applied to the design, screening, and production of growth factor and cytokine mimetics.

[1] Research Institute for Bioscience Products & Fine Chemicals, Ajinomoto Co., Inc., 1-1, Suzuki-Cho, Kawasaki-ku, Kawasaki-shi, Kanagawa 210-8681, Japan.
✉email: kenichiro.ito.qf8@asv.ajinomoto.com

Growth factors and cytokines are of many types and are involved in the development of organisms by differentially regulating functions such as cell proliferation and differentiation[1,2]. In recent years, the demand for growth factors and cytokines has been increasing with the increase in the research and development of regenerative medicine and stem cell culture. However, several growth factors and cytokines as well as proteins in general still have issues in terms of stability[3] and manufacturing cost. Mimetics of growth factors and cytokines are expected to solve these issues, and mimetics using peptide binders and nucleic acid aptamers are being developed[4–11]. Especially, macrocyclic peptides are attractive binders for these mimetics because their selection methods have been established and they usually show high affinity and selectivity toward target proteins[12–14].

Macrocyclic peptide binders have been successfully turned into growth factor and cytokine mimetics by dimerization[4–6] (Fig. 1a). These mimetics activate endogenous receptors via receptor dimerization, evoke downstream cellular signalings[15], and promote changes in cellular morphologies in a similar manner to natural ligands, both in vitro and in vivo. These achievements have made multivalent macrocyclic peptides attractive as novel growth factor and cytokine mimetics.

To date, various chemical and biological methods of producing multivalent macrocyclic peptides have been reported. Among them, the most used strategy is chemical conjugation of synthesized peptides with chemical linkers[16–18]. This method can produce both the homodimers and heterodimers of various combinations but requires multiple synthesis and purification steps. Chemical synthesis using branched amino acids or resins has been reported[18,19]. This method has the advantage of omitting the conjugation step, but heterodimer synthesis is considered difficult. Alternatively, biological synthesis methods using microorganisms have been reported[20]. However, the bacterial expression of relatively small multivalent peptides requires protein tags and cleavage steps. Therefore, a versatile and efficient method for producing multivalent macrocyclic peptides would be a powerful tool for the development of growth factor and cytokine mimetics.

The present work reports the use of single-chain tandem macrocyclic peptides (STaMPtides), a multivalent peptide format compatible with the secretory production method of *Corynebacterium glutamicum*[21]. Highly active growth factor and cytokine mimetic STaMPtides were designed based on previously reported macrocyclic peptide binders and successfully secreted by *C. glutamicum*. The secreted STaMPtides contained two or more macrocyclic moieties with disulfide bonds and peptide linkers of 8–200 amino acids, as designed. The STaMPtides also showed high agonistic activities in cell-based evaluations in both decellularized supernatant and purified forms. The results demonstrated the applicability of STaMPtide technology to the design, screening, and production of growth factor and cytokine mimetics for cell culture and regenerative therapy.

## Results

**Design and expression scheme of STaMPtides.** The molecular format of STaMPtides was designed as a single-chain (Fig. 1b) so that they can be biologically expressed. STaMPtides consisted of multiple macrocyclic peptides closed by disulfide bonds between cysteines, the C- and N-termini of which were linked by a peptide linker. Expression vectors encoding STaMPtides were transformed into *C. glutamicum* for secretion, as previously reported[21],

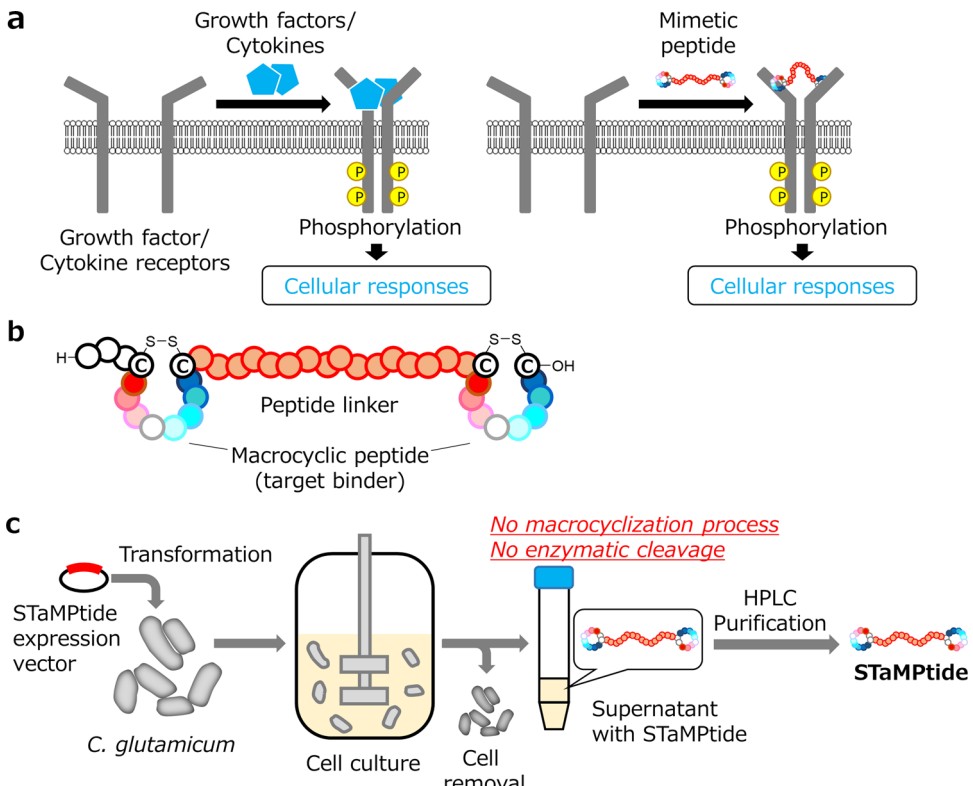

**Fig. 1 Overview of the research. a** Activation mechanism of cytokine and growth factor receptors by their ligands. Growth factors, cytokines, and their mimetics bind to their receptors to form heterocomplexes. Cellular signals are evoked by autophosphorylation of the receptors. **b** Structure of single-chain tandem macrocyclic peptides (STaMPtides), which include multiple disulfide-closed macrocyclic peptide binders linked by peptide linkers. **c** STaMPtide production. STaMPtides are secreted by transformed *Corynebacterium glutamicum* into a supernatant and purified by decellularization and conventional HPLC.

and supernatants were obtained for the analysis of the expression results and preliminary evaluation of the activities of STaMPtides without macrocyclization or enzymatic cleavage (Fig. 1c). To obtain purified STaMPtides, conventional purification by high-performance liquid chromatography (HPLC) was performed.

**Expression of hepatocyte growth factor mimetic STaMPtides.** We designed and expressed STaMPtides with a peptide sequence of a macrocyclic peptide aMD4, which was discovered as a binder against human Met, a receptor of hepatocyte growth factor (HGF)[6]. Since dimers of aMD4 activate endogenous human Met in the same manner as canonical HGF, it is suitable to show the applicability of STaMPtides to artificial growth factor mimetics. Because aMD4 contains a noncanonical amino acid and a macrocyclic structure[6], we needed to slightly modify its structure to make it biologically expressible. Specifically, a modified macrocyclic peptide aMD4dY was designed by elimination of an N-terminal noncanonical amino acid (D-Tyr) of aMD4 and replacement of an intramolecular thioether linkage by a disulfide linkage (Supplementary Table 1). Despite the structural replacement, aMD4dY retained its binding capacity to human Met, as indicated by surface plasmon resonance (SPR) analysis (Supplementary Fig. 1a and Supplementary Table 2).

To develop HGF mimetic STaMPtides, two aMD4dY macrocyclic peptide moieties were designed to be linked by conventional Gly-Ser (GS) and Pro-Ala (PA) tandem repeats of various lengths to determine optimal expression results and HGF mimetic activity (Fig. 2a and Supplementary Table 1). Pro-Ala-tandem sequences were selected because they are highly resistant to proteases and slightly immunogenic and form a random coil structure as well as conventional PEG linkers[22,23]. Expression vectors encoding the HGF mimetic STaMPtides designed were transformed and expressed by *C. glutamicum*, as reported[21], and supernatants

were analyzed by sodium dodecyl sulfate–polyacrylamide gel electrophoresis (SDS-PAGE) after decellularization (Fig. 2b). HGF mimetic STaMPtides using PA linkers could be expressed with high efficiency (Supplementary Table 3) from short linkers (8 amino acids) to long linkers (200 amino acids) in a low background of host cell proteins, while those using GS linkers had relatively low efficiency. This result indicated that PA linkers are suitable as peptide linkers for STaMPtides in terms of expression. However, the bands considered to be STaMPtides were detected on the high-molecular-weight (MW) sides of the SDS-PAGE gel rather than the expected MWs because of the large size-exclusion volume of PA linkers[22]. Thus, we analyzed the molecular masses of the expressed STaMPtides with PA linkers to ensure that the STaMPtides were accurately secreted (Supplementary Fig. 2a–e). The detection of expected masses indicated that the STaMPtides were expressed as designed.

It is conceivable that STaMPtides may form multiple patterns of disulfide bonds. Since it is theoretically possible for dimer-type STaMPtides to form three patterns of disulfide bonds (Supplementary Fig. 3), we analyzed the patterns of disulfide bonds of aMD4dY-PA22 and aMD4dY-PA49. Two STaMPtides were digested by thermolysin[24] and ProAlanase[25], respectively, without reduction, and the molecular masses of digested fragments were subsequently analyzed using liquid chromatography–tandem mass spectrometry (LC–MS/MS). The molecular masses of the fragments resulting from the enzymatic digestion of the STaMPtides with the desired disulfide pattern were detected (Fig. 2c, Supplementary Fig. 4a, b). Fragments containing the entire structure of disulfide-closed aMD4dY cleaved from PA linkers were detected via ProAlanase digestion. In the extracted ion chromatogram (XIC) of aMD4dY-PA22 and aMD4dY-PA49 digestives, no peaks derived from peptides with incorrect disulfide patterns were detected, whereas peptide fragments

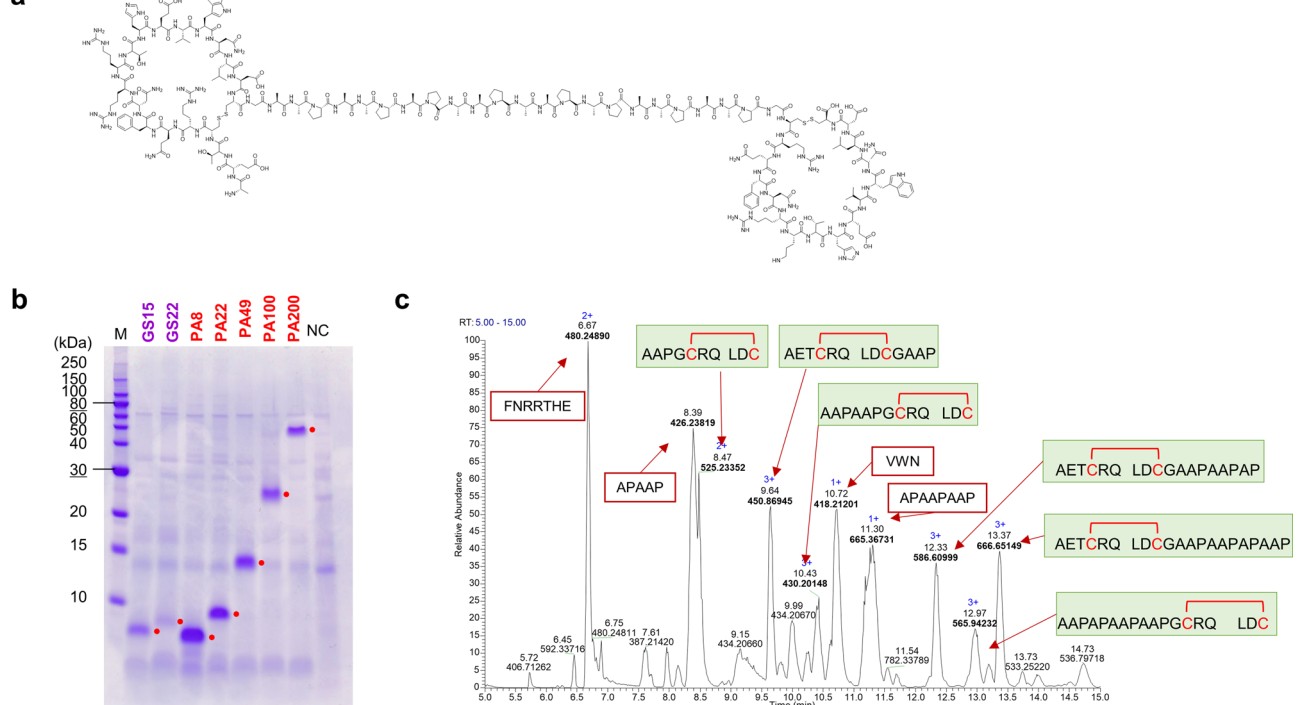

**Fig. 2 Production of HGF mimetic STaMPtides. a** Chemical structure of HGF mimetic aMD4dY-PA22. **b** SDS-PAGE analysis of secreted linker variants of HGF mimetic STaMPtides. Red dots indicate STaMPtide-derived bands. M: marker, NC: negative control supernatant (mock-vector-transfected). **c** Identification of disulfide bonds of aMD4dY-PA22. aMD4dY-PA22 was submitted for thermolysin digestion without reductive alkylation, and peptide fragments were analyzed by LC–MS/MS. The base peak chromatogram of 5–15 min is shown, and peaks of disulfide-containing fragments are highlighted in green boxes.

from the expected disulfide pattern were detected (Supplementary Fig. 5a–c). These results clearly indicated that STaMPtides secreted by *C. glutamicum* contain macrocyclic structures with correctly formed intramolecular disulfide bonds.

**Bioactivity of HGF mimetic STaMPtides**. To identify the optimal linker for agonistic activity against Met, we assessed the activity of STaMPtides without purification beyond decellularization using the HGF-responsible extracellular signal-regulated protein kinase (Erk) reporter assay (Supplementary Fig. 6a). This rapid activity-based screening was made possible because STaMPtides can be secreted directly into the low cross-reactive supernatant in active form (Supplementary Fig. 6b, c), rather than being a conventional inactive inclusion body expression. Reporter assay revealed that all linker variants of HGF mimetic STaMPtides stimulated Erk signaling with previously reported bell-shaped dose dependency[6,15]. Among the lengths of 8–200 amino acids, the HGF mimetic STaMPtide with a 22-amino-acid–PA linker (aMD4dY-PA22) was optimal with the highest activity (Supplementary Fig. 6b). STaMPtides with too long or too short linkers were suggested to have low activity because they caused Met dimerization at inappropriate distances, as previously reported[6]. The HGF mimetic trimer STaMPtide was also designed and successfully expressed (Supplementary Figs. 2f and 7), with the expectation of higher activity than the dimer. However, it agonized Met to the same extent as aMD4dY-PA22 (Supplementary Fig. 6d). For further bioactivity evaluation of HGF mimetic STaMPtides, aMD4dY-PA22 was purified by conventional reverse-phase HPLC (Supplementary Figs. 8a and 9a).

In contrast to symmetrical HGF mimetic peptide dimers reported in previous studies[6,15], aMD4dY-PA22 has an asymmetrical structure in which a linker is attached to the N- or C-terminus of two macrocyclic peptides in a peptide chain. As peptide modification and the disruption of symmetry lead to a loss of peptide function, we performed a detailed functional evaluation of asymmetric aMD4dY-PA22. First, the binding ability of aMD4dY-PA22 to the human Met ectodomain was evaluated using SPR (Supplementary Fig. 1b). The dissociation constant ($K_D$) of aMD4dY-PA22 with human Met was 1.0 nM (Supplementary Table 2), which was 45 times lower than that of monomeric aMD4dY and was consistent with previously reported bivalent molecules (e.g., peptides, fragment antibodies, antibodies)[19,26–31]. The stronger binding of STaMP-tides was attributed to the improved dissociation rate ($k_d$) caused by avidity by bivalency, as previously reported[6,19]. These results indicated that the binding potency of aMD4dY to the Met ectodomain is retained even after terminal modification and is further enhanced by dimerization.

The cellular Met-signaling activation of aMD4dY-PA22 was evaluated using phospho-Met ELISA[6,32] and the Erk signaling reporter assay compared with recombinant human HGF (rhHGF). Of note, aMD4dY-PA22 promoted Met phosphorylation and activated human Erk-serum response element (SRE) signaling, which is a downstream pathway of Met, at a comparable level with rhHGF, although it required a higher molar concentration, whereas monomeric aMD4dY did not activate Met (Fig. 3a, b). To assess the selectivity of aMD4dY-PA22 against RTKs, the lysates of stimulated and unstimulated HuCCT1 cells were analyzed using phospho-RTK array. Met was phosphorylated, whereas other RTKs remained intact, indicating that aMD4dY-PA22 and rhHGF are highly Met-selective (Fig. 3c). This suggests that Erk signaling activation is triggered by Met-selective activation.

Cellular signaling evoked by a Met agonist induces a variety of morphological changes in stimulated cells, including migration, proliferation, and wound healing[6,15], but the extent of these

changes differs between full and partial agonists[32]. Thus, we examined the agonistic potency of aMD4dY-PA22 using the human cell line HuCCT1, which is HGF responsive with all the morphologies mentioned before. First, the effect of aMD4dY-PA22 on the proliferation of HuCCT1 cells was evaluated, which is one of the most fundamental cellular responses to HGF stimulation. HuCCT1 cells were cultured for 5 days with or without rhHGF or aMD4dY-PA22 as a stimulant, and stimulation with rhHGF and aMD4dY-PA22, respectively, promoted twofold proliferation compared to the unstimulated condition (Supplementary Fig. 10a). Second, the effect of aMD4dY-PA22 on migration activity was assessed as another morphological change using Transwell assay. aMD4dY-PA22 and rhHGF also promoted migration of HuCCT1 cells (Supplementary Fig. 11a and b). Since aMD4dY-PA22 promoted the most fundamental cellular morphological changes, we evaluated the effects of aMD4dY-PA22 on wound healing and branching morphogenesis, which are more complex morphological changes. The wound healing rate was assessed by time-lapse imaging of scratched monolayers of HuCCT1 cells. In the presence of rhHGF, HuCCT1 cells showed 1.6-fold faster wound healing than those without stimulation. aMD4dY-PA22 also increased wound healing by 1.8-fold, to the same extent as rhHGF (Fig. 3d). Finally, the effect of aMD4dY-PA22 on branching morphology, the most dynamic structural changes in cells, was investigated. HuCCT1 cells were encapsulated in collagen gels and continuously stimulated by rhHGF or aMD4dY-PA22. rhHGF- or aMD4dY-PA22-stimulated cells showed branched tubular structures after 7 days of incubation, whereas unstimulated cells formed spherical structures (Fig. 3e). Taken together, these morphological changes were induced by aMD4dY-PA22 as well as rhHGF, suggesting that aMD4dY-PA22 is likely a full Met agonist rather than a partial one.

One of the applications of HGF is as a medium supplement for human mesenchymal stem cell (hMSC) culture[33]. To investigate the applicability of STaMPtides to stem cell culture, the effect of stimulation with aMD4dY-PA22 on the growth of bone-marrow-derived hMSCs (BM-hMSCs) was examined compared with rhHGF. aMD4dY-PA22 and rhHGF slightly promoted proliferation and changed the morphology of BM-hMSCs compared with unstimulated cells (Supplementary Fig. 10b, c), which was consistent with previous research[33]. To confirm that stimulated cells maintained the characteristics of MSCs, we further analyzed the expression of surface markers[34,35] and cytokine secretome, which are related to the efficacy of MSCs. BM-hMSCs stimulated with rhHGF or aMD4dY-PA22 expressed hMSC positive markers (CD105, CD73, CD90) and no negative markers (CD45, CD34), similar to unstimulated parental BM-hMSCs, indicating that these cells were undifferentiated (Supplementary Fig. 12a, b). In addition, the supernatant conditioned by rhHGF or aMD4dY-PA22 stimulated BM-hMSCs that contained cytokines and growth factors (IL-6, IL-8, CCL2, angiogenin, VEGF-A, IGFPB-2, osteoprotegerin, TIMP-1, and TIMP-2) to the same extent as unstimulated parental BM-hMSCs (Supplementary Fig. 12c, d). As there was no significant change in the marker expression pattern and cytokine secretome occurred through stimulation, aMD4dY-PA22 promoted the growth of BM-hMSC without a loss of characteristics. These data demonstrate the potential of STaMPtides for utilization as a chemically defined supplement for stem cell culture.

**Development of erythropoietin and thrombopoietin mimetic STaMPtides**. To assess the applicability of our strategy for the development of growth factor and cytokine mimetics, we designed STaMPtides that mimic erythropoietin (EPO) and thrombopoietin (TPO) as previously reported[5,36,37]. A disulfide-cyclized macrocyclic

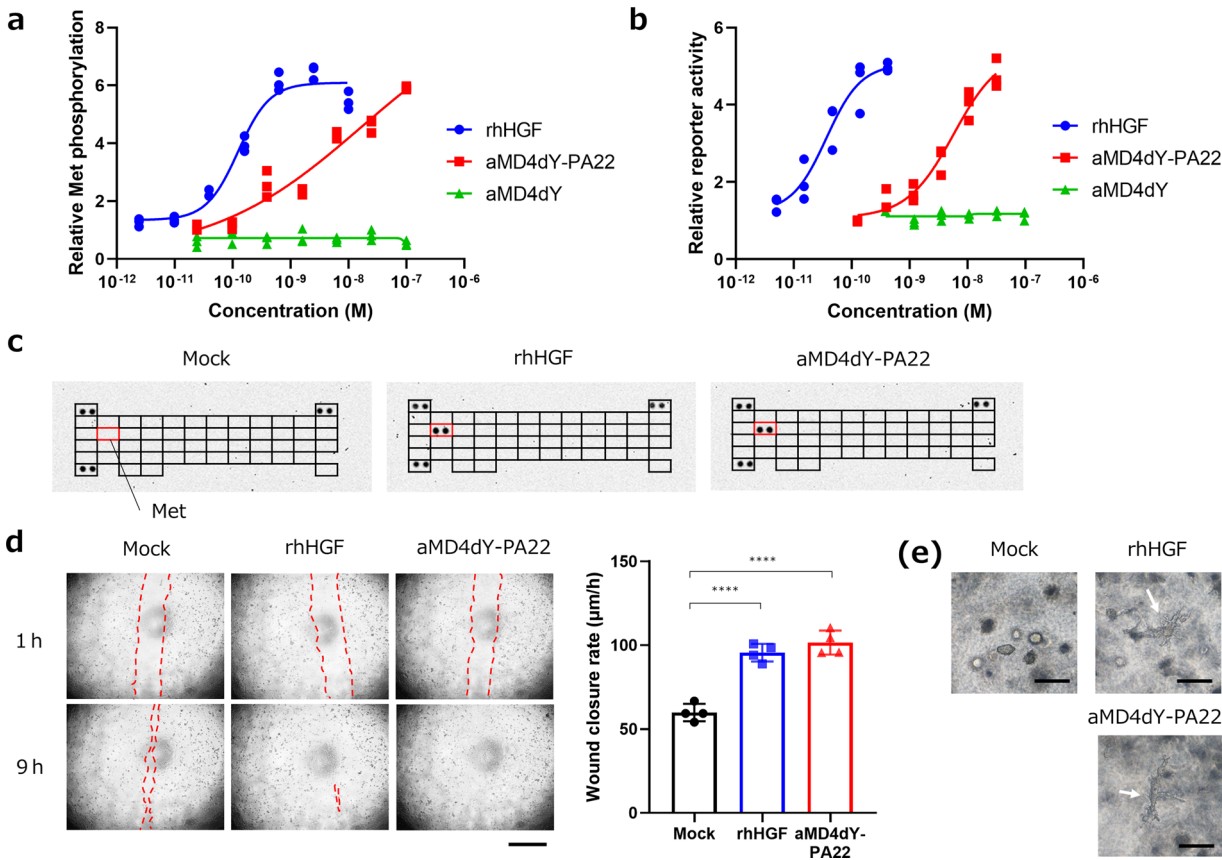

**Fig. 3 Bioactivity evaluation of aMD4dY-PA22. a** Phosphorylation of Met induced by aMD4dY-PA22. Met phosphorylation (Y1234/Y1235) level of HuCCT1 cells stimulated using aMD4dY-PA22 (red squares), rhHGF (blue circles), and monomeric aMD4dY (green triangles) was evaluated via phosphor-Met ELISA. The mean and individual values are shown from the results of triplicate experiments. **b** Signaling potencies induced by aMD4dY-PA22. Erk signaling activation by aMD4dY-PA22 (red squares), rhHGF (blue circles), and monomeric aMD4dY (green triangles) was evaluated by Erk-SRE reporter assay. The mean and individual values are shown from the results of triplicate experiments. **c** Selectivity of aMD4dY-PA22 against various RTKs. Cell lysates of HEK293 cells stimulated by 1.3 nM rhHGF or 32 nM aMD4dY for 10 min were analyzed by a phospho-RTK array. Positions of phospho-Met are indicated by red boxes. **d** Wound healing promoted by aMD4dY-PA22. Wound closure events of HuCCT-1 cell monolayer at 1 and 9 h after scratching with or without 0.41 nM rhHGF or 16 nM aMD4dY-PA22 were captured. Scratches are depicted by red broken lines. Scale bars = 1 mm. Average wound closure rates from 1 to 7 h of each experimental group were calculated. The mean ± SD with individual values are shown from the results of quadruplicate experiments. ****$p < 0.0001$. **e** Branching morphogenesis induced by aMD4dY-PA22. HuCCT-1 cells were encapsulated in collagen gels and cultured in the presence of 0.44 nM rhHGF or 6.3 nM aMD4dY-PA22 for 7 days. Arrows indicate tubule structures. Scale bars = 200 μm.

peptide (EMP35)[36] that binds to and activates the EPO receptor (EPOR) was dimerized as a single-chain peptide with 8- and 22-amino-acid–PA linkers to engineer EMP-PA8 and EMP-PA22 (Supplementary Fig. 13a, b). Likewise, a TPO receptor (MPL)-binding macrocyclic peptide (TMP)[5,37] was dimerized with a PA linker with eight amino acids to form TMP-PA8 (Supplementary Fig. 13c). All newly designed EPO and TPO mimetic STaMPtides were tested for expression using the same *C. glutamicum*-based secretion method utilized for HGF mimetic STaMPtides. The SDS-PAGE (Supplementary Fig. 14a, b) and LC–MS (Supplementary Fig. 2g–i) analyses of decellularized supernatants suggested that STaMPtides are secreted into the supernatant with the correct peptide sequences.

The bioactivities of EPO mimetic STaMPtides were rapidly evaluated using a decellularized supernatant containing each STaMPtide. The β-galactosidase assay reconstituted in response to Janus kinase 2 (JAK2) phosphorylation induced by EPOR activation (Supplementary Fig. 15a) was used to evaluate EPOR-specific agonist activities of EMP-PA8 and EMP-PA22. Serially diluted supernatants including EMP-PA8 or EMP-PA22 were added to reporter cells, and their reporter activity was quantified (Supplementary Fig. 15b). Both EMP-PA8 and EMP-PA22

activated EPOR, even at high dilutions, suggesting that both STaMPtides are highly active. Since EPO mimetic STaMPtides were confirmed to have optimal agonistic activities, they were purified by HPLC for further bioactivity evaluations (Supplementary Figs. 8b, c and 9b, c).

The kinetic constants of two EPO mimetic STaMPtides, EMP-PA8, EMP-PA22, and monomeric EMP35 against recombinant EPOR ectodomain were evaluated via SPR (Supplementary Fig. 1c–e). All the constructs showed strong binding potency against EPOR. Although exact dissociation constants could not be determined because of their too slow dissociation, the association rate of EMP-PA8 and EMP-PA22 was revealed to be 5–10 times higher than that of EMP35 (Supplementary Table 2). Then, the EPOR-activating ability of EMP-PA8 and EMP-PA22 was compared with that of recombinant human EPO (rhEPO) and the monomeric EMP35 peptide using the EPOR-JAK2 functional reporter assay. The results showed that the half-maximal effective concentration ($EC_{50}$) of EMP-PA8 and EMP-PA22 was 22 pM and 220 pM, respectively, which indicated that they are as potent as rhEPO ($EC_{50} = 45$ pM) (Fig. 4a). However, EMP35 was a weak agonist, ~1000 times weaker than STaMPtides, with an $EC_{50}$ of 100 nM, thereby demonstrating an enhancement of potency by

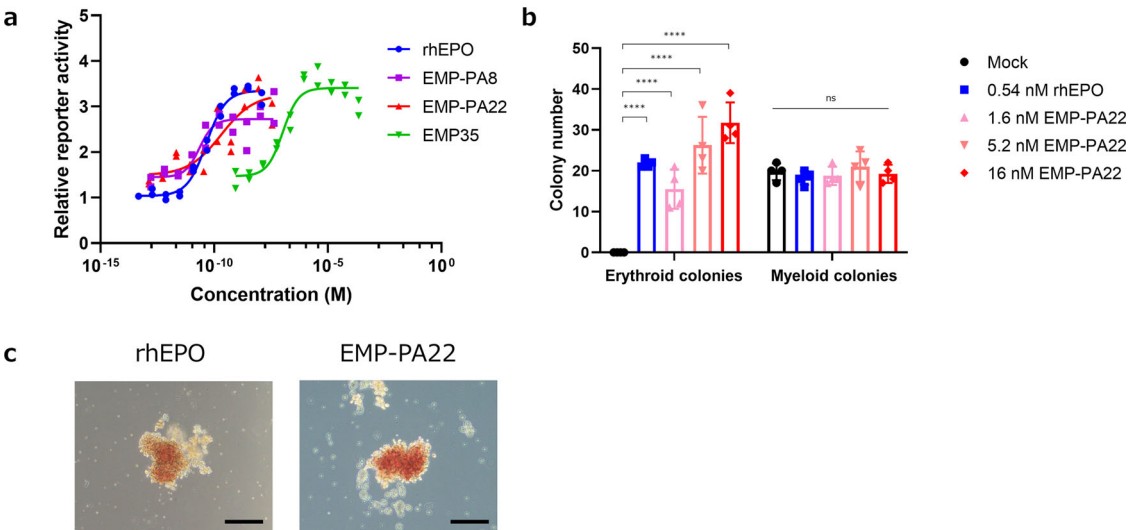

**Fig. 4 Bioactivity evaluation of EPO mimetic STaMPtides. a** Signaling potencies induced by EMP-PA8 and EMP-PA22. EPOR-JAK2 signaling activation by EMP-PA8 (purple squares), EMP-PA22 (red triangles), rhEPO (blue circles), and monomeric EMP35 (green inverted triangles) was evaluated by EPOR-JAK2 functional reporter assay. The mean and individual values are shown. **b** Colony-forming cell assay. hPBMCs were cultured in methylcellulose culture medium in the absence (black bar) or presence (blue bar) of 0.54 nM rhEPO or 1.6–16 nM EMP-PA22 (red bars). Number of erythroid and myeloid colonies formed from $2 \times 10^5$ hPBMCs were counted. The mean ± SD with individual values are shown from the results of quadruplicate experiments. ****$p < 0.0001$. **c** Morphology of erythroid colonies formed in methylcellulose medium in the presence of 0.54 nM rhEPO or 16 nM EMP-PA22 after 13 days of culture. Scale bars = 200 μm.

dimerization, as previously reported[36]. These results suggested that the dimerization of the EPO mimetic macrocycle by a peptide linker reduces the entropic disadvantage, as previously discussed[36,38], and contributes to improved potency.

To demonstrate the applicability of EPO mimetic STaMPtides to stem cell culture, we evaluated the effect of EMP-PA22 on colony formation of hematopoietic progenitor cells. Since hematopoietic progenitor cells differentiate into erythroid cells with EPO stimulation[2], we considered whether EPO mimetic STaMPtides would work as an alternative for rhEPO. Human peripheral blood mononuclear cells (hPBMCs) were cultured in a methylcellulose-based semisolid medium with or without rhEPO or EMP-PA22, and the colony-forming efficiencies of erythroid-progenitor cells (BFU-E and CFU-E) and myeloid progenitor cells (CFU-GM) were compared. Myeloid colonies were formed regardless of the presence or absence of rhEPO stimulation (Supplementary Fig. 16a), but erythroid colonies were formed only in the presence of rhEPO stimulation (Supplementary Fig. 16b). Depending on the dosage of EMP-PA22, erythroid colony formation was promoted, and the formation efficiency was equivalent to or higher than that of 10 ng/mL of rhEPO at doses of 33 ng/mL or higher (Fig. 4b). Moreover, many reddish colonies were formed with stimulation by 33–100 ng/mL of EMP-PA22 (Fig. 4c and Supplementary Fig. 16b), indicating that EMP-PA22 can promote colony formation and maturation of erythroid-related cells at the same level as rhEPO.

Next, the TPO-like function of TMP-PA8 was evaluated. We compared the trend of signals activated by recombinant human TPO (rhTPO) and the *C. glutamicum* supernatant containing TMP-PA8 using a phosphokinase array. Analysis of cell lysates from TPO-responsive human erythroleukemia HEL cells[39] stimulated with rhTPO or TMP-PA8-containing supernatants by phosphokinase array revealed that phosphorylation of signal transducer and activator of transcription (STAT)3, STAT5, Erk1/2, and adenosine 3′,5′-cyclic monophosphate (cAMP)-responsive element-binding protein (CREB) is enhanced in rhTPO- and TMP-PA8-stimulated HEL cells compared with unstimulated

cells (Supplementary Figs. 17a, b). This result is consistent with previous reports[40,41] that TPO-MPL activates JAK2 and subsequent STAT3/5 and Erk1/2 pathways, suggesting that TMP-PA8 is a TPO-like receptor agonist.

TMP-PA8 was purified (Supplementary Figs. 8d and 9d), and its comparative activity was quantified by assessing the phosphorylation levels of the CREB transcription factor using phospho-CREB-specific enzyme-linked immunosorbent assay (ELISA) compared with rhTPO. CREB was chosen as a marker for TPO activity because the promotion of CREB phosphorylation by TPO-mediated signaling has been previously reported and was also confirmed by our TPO signaling analysis (Supplementary Fig. 17a, b). HEL cells were stimulated with rhTPO or HPLC-purified TMP-PA8, and their phosphorylation levels were quantified by phospho-CREB ELISA. The results showed that both rhTPO and TMP-PA8 promote CREB phosphorylation by 1.5-fold at concentrations of 0.1–1 nM (Fig. 5), suggesting that TMP-PA8 acts as an MPL agonist. The results of the design, expression, and bioactivity evaluation of three STaMPtide-based growth factor mimetics (HGF, EPO, and TPO mimetics) indicates that STaMPtides are a versatile method for developing potent growth factor mimetics.

## Discussion

A variety of multimeric macrocyclic peptides have been developed as molecules with novel functions. It is possible to obtain peptide binders by well-established methods, as exemplified by phage display and messenger RNA (mRNA) display. However, conventional methods of producing multivalent macrocyclic peptides involve complex steps, such as chemical linker conjugation, branching, and macrocyclization[16–19] in the case of chemical synthesis and post-expression refolding and enzymatic treatment[20] in the case of biosynthesis, which are disadvantageous for yield, scalability, and discovery throughput. By combining STaMPtides, which is a multivalent peptide format consisting of multiple macrocyclic peptides and peptide linkers, with a unique secretion technology using *C. glutamicum*[21], this

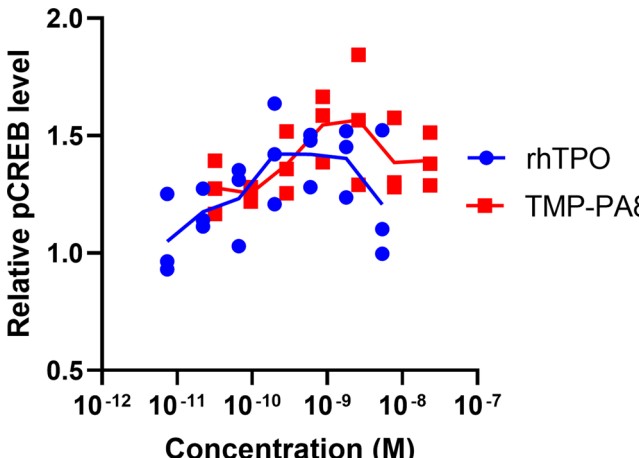

**Fig. 5 Bioactivity evaluation of the TPO mimetic STaMPtide.** Signaling potencies induced by TMP-PA8. Starved HEL cells were stimulated by TMP-PA8 (red squares) or rhTPO (blue circles) for 10 min. Relative phosphorylation levels of CREB were evaluated by phospho-CREB ELISA. The mean and individual values are shown from the results of triplicate experiments.

study overcame these challenges and developed a methodology for developing growth factor and cytokine mimics. Although STaMPtide may be applicable to other expression systems, the *C. glutamicum*-based expression system should be more suitable because it efficiently secretes STaMPtides into the supernatant in an active form with correct folding, including disulfide bonds, whereas some hosts (i.e., *E. coli*) may express them as inclusion bodies in the cell or require expression tags/cargoes to prevent degradation[20]. As shown by disulfide analysis, the majority of the secreted STaMPtides formed multiple disulfide bonds intramolecularly as designed and high yields were obtained since no additional steps were required. Furthermore, *C. glutamicum* hardly secretes host cell proteins into the supernatant nor produces endotoxin natively; hence, the target STaMPtide can be secreted endotoxin-free with high purity. These features enable the rapid engineering of STaMPtides, which is a main advantage of this technology. STaMPtides can be produced by general vectors and expression work, and cell assays can be performed using supernatants containing STaMPtides. In fact, the optimal linker length of the HGF mimetic STaMPtide was rapidly identified by cellular assay using a supernatant. Therefore, STaMPtide technology can be a powerful tool for the rapid development of molecules, such as growth factor and cytokine mimetics, in which the linker length greatly affects the activity.

The main concern with STaMPtides compared with previous reports is their asymmetry. Most growth factors, cytokines, and their receptors are or form symmetric homodimers, and their complexes are usually also symmetric. Since STaMPtides are asymmetric due to their design of connecting C- and N-termini, it is unclear whether they can activate their receptors. Paradoxically, STaMPtides are biologically active, suggesting that they can form active symmetric complexes with their receptors. To clarify whether symmetry is necessary, molecular dynamics (MD) simulations were performed to predict the structure of the STaMPtide–target receptor complex. The structure of the EMP-PA22–EPOR complex was predicted by homology modeling and compared with that of the EPO-binding peptide EMP1–EPOR complex (Supplementary Table 1), whose crystal structure has already been unraveled[38]. As a control for the symmetric EPOR–agonist peptide dimer, we used a peginesatide core, which is a peptide dimer domain of peginesatide[42] (Supplementary Table 1), which was originally developed as an alternative to EPO.

MD simulation of the complexes of EPOR and these three peptide ligands was performed to predict whether they could be maintained. Analysis of the receptor orientation in the stable state assessed by root-mean-square deviation (Supplementary Fig. 18a) and predicted structures revealed that the homodimer of EPOR quickly disintegrates in the absence of the peptide ligand (Supplementary Fig. 18b) but remains stable with EMP1 during 100 nm simulation (Supplementary Fig. 18c). The peginesatide core and EMP-PA22 maintain the homodimer of EPOR as a structure similar to the complex with EMP1 (Supplementary Fig. 18d, e). Changes in the distance and angle between two receptors change the nature and intensity of the activated signal[43]. Since these values are close to each other, EMP-PA22 is expected to exhibit similar activity to the highly symmetric peginesatide core, even though it is asymmetric. One reason growth factor and cytokine mimetics do not require perfect symmetry is that the N- and C-termini of the macrocyclic peptide moieties are expected to be located opposite to the receptor-binding position and spatially close to each other. In fact, both N- and C-termini of the EPOR-binding macrocyclic peptide are located spatially close to each other on the opposite side of EPOR in the EMP1 crystal structure[38] and in the MD simulation results of the peginesatide core and EMP-PA22 (Supplementary Fig. 18f, g). In addition to EMP1, the crystal structures of macrocyclic peptides discovered by the peptide display method and their target receptors have been analyzed[44,45]. These analyses show that the N- and C-termini of cyclic peptides tend to face the opposite direction of the binding site of the target proteins. In the peptide display method, peptides are pulled down by the target with a large molecule, such as a phage virus or a nucleic acid, bound to their terminus[12]. Therefore, it is expected that the sterically hindered end tends to face outward. Because the artificial macrocyclic peptide conjugates obtained by the peptide display method can be modified at the termini, they are highly compatible with STaMPtide development.

Since cell morphology and fate are regulated by a complex balance of intracellular signals, alternatives to growth factors and cytokines should work as close as possible to their canonical ligands and be full agonists, especially to be used in stem cell culture. As shown in various cellular experiments, STaMPtides with a precisely controlled linker length may work as growth factor or cytokine mimetics comparable to natural growth factors and cytokines, in terms of both maximal activity and induction of phenotypic changes in cells. In stem cell-related technologies, a large variety of growth factors and cytokines are used to control the fate and growth of stem cells. Therefore, further discovery of their mimetics based on STaMPtides is underway. To develop mimetics of many kinds of growth factors and cytokines, it is desirable to develop library construction and screening methods to screen active dimers and multimers in a more high-throughput manner, and such methods are also currently under development.

STaMPtide technology is highly versatile, and expression of not only homodimers but also heterodimers and multimers is under investigation. In recent years, there have been attempts to enhance the functionality of biologics by adding functional peptide sequences such as blood-stable[46] and matrix-binding[47] peptide sequences. STaMPtides can be added with these functional peptides in tandem to be functionalized. We believe that in the future, such strategies will enable us to not only improve the properties of mimetics presented in this study but also to develop a variety of functional molecules.

## Methods
**Materials**. Recombinant human Met ectodomain-Fc, rhEPOR ectodomain-Fc, and rhHGF were purchased from R&D Systems (Minneapolis, MN, USA). rhEPO and rhTPO were purchased from Peprotech, Inc. (Rocky Hill, NJ, USA). Monomeric

macrocyclic peptides (aMD4dY and EMP35) were synthesized by GenScript Japan (Tokyo, Japan). Ammonium acetate was purchased from Nacalai Tesque (Kyoto, Japan). Thermolysin and ProAlanase were purchased from Promega Corporation (Madison, WI, USA). Trifluoroacetic acid (TFA), formic acid, and acetonitrile were purchased from FUJIFILM Wako Pure Chemical Corporation (Osaka, Japan).

**Cell culture**. HEK293 cells were purchased from the American Type Culture Collection (Manassas, VA, USA). HuCCT1 and HEL cells were purchased from the Japanese Cancer Research Resources Bank (Japan). BM-hMSCs and hPBMCs were purchased from Lonza Group AG (Switzerland). HEK293 cells were maintained in Dulbecco's modified Eagle medium (DMEM) high-glucose GlutaMAX supplement (Thermo Fisher Scientific, Waltham, MA, USA) supplemented with 10% fetal bovine serum (FBS; Thermo Fisher Scientific) and 1% penicillin-streptomycin (Nacalai Tesque). HuCCT1 and HEL cells were maintained in RPMI 1640 medium (Nacalai Tesque) supplemented with 10% FBS and 1% penicillin-streptomycin. BM-hMSCs were maintained in DMEM (Thermo Fisher Scientific) supplemented with 10% FBS and penicillin-streptomycin.

***C. glutamicum* expression and purification of STaMPtides**. STaMPtides were expressed by *C. glutamicum*, as previously described. Briefly, STaMPtides with the *C. glutamicum* ATCC13869-derived CspB signal peptide were encoded downstream of the ATCC13869-derived *cspB* gene promoter in the pPK4 expression vector[48]. The constructed expression vectors were transformed into the *C. glutamicum* YPS010 strain[49]. The transformed cells were cultured in MMTG liquid medium supplemented with 25 mg/L of kanamycin at 30 °C for 72 h. Next, 6.5 µL of the supernatant was analyzed by reduced SDS-PAGE using NuPAGE 12% Bis-Tris Gel (Thermo Fisher Scientific) and Quick-CBB (Wako Pure Chemical Industries). The supernatant was decellularized by filtering through an SLGV033RS Hydrophilic PVDF Millex-GV Filter Unit (Merck Millipore, Burlington, MA, USA).

Decellularized STaMPtides were purified by conventional reverse-phase HPLC (RP-HPLC) using the Alliance e2695 HPLC System (Waters Corporation, Milford, MA, USA) under linear gradient conditions with mobile phases A (water with 0.1% TFA) and B (80% acetonitrile, 19.9% water, and 0.1% TFA in acetonitrile).

**LC-MS analysis of STaMPtides**. The molecular masses and yield of STaMPtides were measured by LC–MS using ACQUITY UPLC H-Class/SQD2 (Waters Corporation) under linear gradient conditions with mobile phases A (water with 0.1% TFA) and B (80% acetonitrile, 19.9% water, and 0.1% TFA in acetonitrile). The yields of STaMPtides were calculated by comparing their peak areas in culture broth concentrated by Amicon Ultra 3 kDa (Merck) five times with that of the 0.2 mg/mL bovine serum albumin standard. All the purified STaMPtides were confirmed to have >95% purity.

**Disulfide bond analysis using thermolysin and ProAlanase**. Each sample of 0.5 µg was diluted with 50 mM ammonium acetate (pH 6.6 for thermolysin digestion) or KCl-HCl (pH 1.5, for ProAlanase digestion) to form 30 µL aliquots. For thermolysin digestion, 200 ng of thermolysin was added and incubated for 3 h at 65 °C. To quench the digestion, 20% TFA was added to the final concentration of 0.5%. For ProAlanase digestion, 20 ng of ProAlanase was added and incubated for 2 h at 37 °C. To quench the digestion, samples were heated up to 95 °C for 10 min. LC–MS/MS analysis was carried out using Easy-nLC 1000 (Thermo Fisher Scientific) connected to Orbitrap Fusion Tribrid (Thermo Fisher Scientific). Acclaim PepMap® 100 (75 µm × 2 cm, Thermo Fischer Scientific), and electrospray ionization (ESI) (75 µm × 12.5 cm, 3 µm, NTCC-360/75-3-125; Nikkyo Technos) columns were used for as trap and analytical columns, respectively. The total 59.5 min gradient program was as follows; 0.5 min hold at 2% mobile phase B (0.1% formic acid in acetonitrile), 49.5 min linear gradient from 2% to 50% mobile phase B, 5.5 min hold at 50% mobile phase B, followed by an increase to 95% mobile phase B in 1 min, and 3 min hold. Mobile phase A was a 0.1% formic acid, and the flow rate was set to 300 nL/min. MS analysis was carried out in data-dependent acquisition mode with full scan from $m/z$ 350–2000. The spray voltage and ion transfer tube temperature were set to 1600 V and 275 °C, respectively. The mass resolution of MS1 was 120,000, and collision-induced dissociation was used to acquire MS2 spectra. The maximum injection time for MS1 and MS2 was set to 50 ms, and the top speed mode with 3 s cycles for MS/MS analysis was selected. The resulting MS/MS data were analyzed using Xcalibur (Thermo Fisher Scientific). MS/MS spectrum data were applied to confirm the amino acid sequence and increase identification accuracy.

**SPR analysis**. Binding constants of the STaMPtides and monomeric macrocyclic peptides to their corresponding receptor ectodomains were determined by SPR analysis using Biacore T200 (Cytiva). HBS-EP + buffer (Cytiva) containing 0.1% dimethyl sulfoxide (DMSO; Sigma-Aldrich, St. Louis, MO, USA) was used as the running buffer. As ligands, recombinant human Met ectodomain-Fc and human EPOR ectodomain-Fc were immobilized to around 1000 response units on CM5 sensor chips using a human antibody capture kit (Cytiva). STaMPtides and monomeric macrocyclic peptides were injected in five concentrations at a flow rate

of 30 µL/min using a single-cycle kinetics method. Binding constants were quantified by analysis of the 1:1 binding model.

**SRE reporter assay for HGF activity evaluation**. The Met-activating ability of HGF mimetic STaMPtides was evaluated by SRE reporter assay using the SRE reporter vector (QIAGEN, Hilden, Germany) according to the manufacturer's instructions. Briefly, 1 µL of the SRE reporter vector was mixed with 0.6 µL of Attractene Transfection Reagent (QIAGEN) in 50 µL of Opti-MEM medium (Thermo Fisher Scientific) to form a transfection complex. Next, HEK293 cells at a density of 40,000 cells/well were seeded on 96-well cell culture plates with the transfection complex and incubated overnight in 5% CO₂ at 37 °C. The cells were starved with Opti-MEM containing 0.5% FBS, 1% non-essential amino acid solution, and penicillin-streptomycin and incubated at 37 °C for 4 h. Next, the cells were stimulated by rhHGF, purified HGF mimetic STaMPtides, monomeric aMD4dY, or decellularized supernatant overnight in a 5% CO₂ incubator at 37 °C. The signal intensity was detected using the dual-luciferase reporter assay system (Promega Corporation) and a Nivo plate reader (PerkinElmer, Inc., Waltham, MA, USA). Reporter activity was calculated as (luminescence intensity of Firefly luciferase)/(luminescence intensity of Renilla luciferase) and normalized by that without a stimulant in order to determine the relative reporter activity.

**Cell-based phospho-Met ELISA**. The phosphorylation level of Met was evaluated according to a previously reported method[6,32], with slight modifications. In brief, HuCCT-1 cells at a density of 10,000 cells/well were seeded on 96-well cell culture plates and incubated overnight in 5% CO₂ at 37 °C. The cells were stimulated by rhHGF, purified HGF mimetic STaMPtides, and monomeric aMD4dY for 10 min in a 5% CO₂ incubator at 37 °C. Next, cells were fixed using 4% paraformaldehyde in phosphate-buffered saline (PBS) for 20 min. Cells were washed three times with PBS and blocked with 1% BSA and 0.02% Triton X-100 in PBS for 30 min at room temperature. Cells were incubated with 1:1000 diluted phospho-Met (Tyr1234/1235) (D26) XP Rabbit mAb (Cell Signaling Technology) supplemented with 1% BSA in PBS at 4 °C overnight. After washing three times with PBS, the cells were incubated with 1:1000 diluted anti-mouse HRP-linked IgG antibody (Cell Signaling Technology) supplemented with 1% BSA in PBS at room temperature for 3 h. Cells were washed four times with PBS and incubated with 100 µL TMB substrate for 30 min. The reaction was stopped with an equal volume of 0.2 N sulfuric acid, and absorbance was quantified at 450 nm by the Molecular Imager ChemiDoc XRS System (Bio-Rad).

**Proliferation assay**. The activity of aMD4dY-PA22 against HuCCT1 cell and BM-hMSC proliferation was evaluated. HuCCT1 cells at a density of 5000 cells/well were seeded on a 24-well cell culture plate with RPMI 1640 medium supplemented with 5% FBS with or without 0.44 nM rhHGF or 6.3 nM aMD4dY-PA22. After 5 days of culture in 5% CO₂ at 37 °C, the cells were detached by 0.25% trypsin/ethylenediaminetetraacetic acid (EDTA; Thermo Fisher Scientific) and counted using a Countess automated cell counter (Thermo Fisher Scientific).

BM-hMSC cells at a density of 2500 cells/well were seeded on a 96-well cell culture plate with DMEM supplemented with 10% FBS with or without 1.3 nM rhHGF or 32 nM aMD4dY-PA22. After 3 days of culture in 5% CO₂ at 37 °C, Cell Counting Reagent SF (Dojindo Laboratories, Japan) was added to the cells. After 1 h incubation, cell numbers were quantified by detection of absorbance at 450 nm using the Benchmark Plus plate reader (Bio-Rad Laboratories, Hercules, CA, USA).

**Flow cytometry**. The marker expression of BM-hMSCs unstimulated or stimulated using 1.3 nM rhHGF or 32 nM aMD4dY-PA22 for 3 days were analyzed via flow cytometry. Harvested cells were blocked with PBS supplemented with Human TruStain FcX (Biolegend) for 10 min on ice. FITC-anti-human CD90 (Biolegend), PE anti-human CD105 (Biolegend), PE anti-human CD34 (Biolegend), FITC-anti-human CD45 (Biolegend), or FITC-anti-CD73 (Biolegend) diluted 10 times with a cell stain buffer (PBS supplemented with 1% FBS) was added to cells and incubated for 1 h on ice. For the isotype controls, 1:10 diluted PE mouse IgG1 k Isotype (Biolegend) and FITC mouse IgG1 k Isotype (Biolegend) were utilized. After washing two times with a cell stain buffer, cells were analyzed using Attune NxT flow cytometer (Thermo Fisher Scientific).

**Cytokine array**. The conditioned media (supernatant) of BM-hMSCs unstimulated or stimulated with 1.3 nM rhHGF or 32 nM aMD4dY-PA22 for 3 days were filtered using Millex-GV 0.22 µm PVDF membrane filter (Millipore) and analyzed using Human Cytokine Array C5 (Ray Biotech) according to the manufacturer's instructions. Chemiluminescence was detected using Amersham Imager 600 (Cytiva).

**Migration assay**. HuCCT1 cells at a density of 20,000 cells/well in RPMI 1640 medium supplemented with 0.5% FBS were added to 24-well Transwell inserts. In the lower layer, RPMI 1640 supplemented with 0.5% FBS with or without 1.3 nM rhHGF or 32 nM aMD4dY-PA22 was added. The cells were incubated for 24 h in 5% CO₂ at 37 °C. The Transwell inserts were washed thrice with PBS, stained with 0.5% crystal violet (Nacalai Tesque)/methanol for 10 min, and washed with

distilled water. Images of migrated cells were captured using BZ-X fluorescence microscopy (Keyence Corporation, Osaka, Japan), and cell migration levels were quantified by stained areas of the Transwell inserts.

**Scratch assay**. HuCCT1 cells at a density of 50,000 cells/well in RPMI 1640 medium supplemented with 10% FBS were added to a 96-well cell culture plate and incubated overnight in 5% $CO_2$ at 37 °C to form cellular monolayers. The centers of the cellular monolayers were scratched with 1000 µL micropipette tips and incubated in RPMI 1640 medium supplemented with 0.5% FBS with or without 0.41 nM HGF or 16 nM aMD4dY-PA22 in 5% $CO_2$ at 37 °C. Images of the cells were captured every hour using BioStudio-T (Nikon, Tokyo, Japan). The average wound healing rate was calculated as [(width of the wound 1 h after scratching) − (width of the wound 6 h after scratching)]/6.

**Branching morphogenesis**. HuCCT1 cells at a density of 20,000 cells/well were mixed with 400 µL of ice-cold collagen-containing culture medium (DMEM/Nutrient Mixture F-12, CellMatrix type IA [Nitta Gelatin Inc., Osaka, Japan], 2.2 g/L of sodium bicarbonate, 4.77 g/L of 4-(2-hydroxyethyl)-1-piperazineethanesulfonic acid [HEPES], and 0.005 N sodium hydroxide) and incubated in a 48-well cell culture plate in 5% $CO_2$ at 37 °C for gelation. The cells encapsulated in the collagen gel were cultured with 500 µL of RPMI 1640 medium supplemented with 10% FBS with or without 0.44 nM HGF or 6.3 nM aMD4dY-PA22 in 5% $CO_2$ at 37 °C. The culture medium was changed every 3 days. At day 7, the cells were observed by inverted microscopy.

**Signal analysis using phosphokinase and phospho-RTK arrays**. Ligand specificity against RTKs was evaluated using the Proteome Profiler Human Phospho-RTK Array Kit (R&D Systems). Confluent HEK293 cells in a 10 cm cell culture dish were starved in DMEM supplemented with 0.5% FBS and penicillin-streptomycin for 24 h. The starved cells were stimulated by 1.3 nM rhHGF or 32 nM aMD4dY-PA22 for 10 min, rinsed once with PBS, and lysed with 1 mL of lysis buffer 17 (R&D Systems) supplemented with a protease inhibitor cocktail (Nacalai Tesque). Lysates were analyzed using the Proteome Profiler Human Phospho-RTK Array Kit (R&D Systems) according to the manufacturer's instructions. Chemiluminescence was detected using the Molecular Imager ChemiDoc XRS System (Bio-Rad Laboratories).

The TPO signal cascade was analyzed using the Proteome Profiler Human Phosphokinase Array Kit (R&D Systems). Briefly, $1 \times 10^6$ HEL cells in a 6-well cell culture dish were starved in RPMI 1640 medium for 24 h. The starved cells were stimulated by 5.4 nM rhTPO or 1:1000 diluted TMP-PA8 containing a *C. glutamicum* supernatant for 20 min, rinsed once with PBS, and lysed with 200 µL of lysis buffer 17 supplemented with a protease inhibitor cocktail. Lysates were analyzed using the Proteome Profiler Human Phosphokinase Array Kit (R&D Systems) according to the manufacturer's instructions. Chemiluminescence was detected using Amersham Imager 600 (Cytiva).

**EPOR-JAK2 functional assay**. Agonistic activity against EPOR was assessed using the PathHunter® eXpress EpoR-JAK2 Functional Assay Kit (DiscoverX) according to the manufacturer's instructions. Briefly, cells were seeded on a 96-well cell culture plate and incubated overnight in 5% $CO_2$ at 37 °C. Serially diluted rhEPO, EPO mimetic STaMPtides, monomeric EMP35 peptide, or a STaMPtide-containing decellularized supernatant was added to the cells and incubated at room temperature for 3 h to stimulate the cells. The cells were lysed with a substrate reagent, and chemiluminescence was detected using a Nivo plate reader (PerkinElmer).

**Colony-forming assay**. Here, 800,000 hPBMCs were suspended in 400 µL of Iscove's modified Dulbecco's medium (Thermo Fisher Scientific) supplemented with 2% FBS and thoroughly mixed with 4 mL of MethoCult™ H4035 Optimum Without EPO (STEMCELL Technologies Inc., Vancouver, Canada) with or without 0.54 nM rhEPO or 1.6–16 nM EMP-PA22. Next, 1.1 mL of the mixture was seeded in a 6-well cell culture plate and cultured for 13 days in 5% $CO_2$ at 37 °C. At day 13, colony numbers were counted, and images were taken by inverted microscopy and BZ-X fluorescence microscopy.

**Phospho-CREB ELISA**. Here, 100,000 HEL cells were seeded on a 96-well cell culture plate in RPMI 1640 medium and incubated overnight for starvation. The starved cells were stimulated by rhTPO or TMP-PA8 for 20 min, rinsed once with PBS, and lysed with lysis buffer 17 supplemented with a protease inhibitor cocktail. The phosphorylation level of CREB was assessed using phospho-CREB (Ser133) and the Total CREB ELISA kit (RayBio) according to the manufacturer's instructions. The amounts of phospho-CREB and total CREB were quantified by measuring absorbance at 450 nm, and the phospho-CREB level was calculated as (absorbance of pCREB)/(absorbance of total CREB).

**Molecular dynamics simulation**. Computational analysis was performed using the Bioluminate software suite version 2020-3 (Schrödinger, Inc., New York, NY, USA). The crystal structure of EPOR was retrieved from the Protein Data Bank (PDB: 1EBP) as a complex with EMP1. Prime software was used to build each homology model of the peginesatide core and EMP-PA22 by using EMP1 as a template[50,51]. The modeling structures of the EPOR–peginesatide core and EPOR-EMP-PA22 were constructed by replacing EMP1 with a peginesatide core and EMP-PA22 in EPOR-EMP1. MD simulations were performed thrice for each structure with an NPT ensemble for 100 ns at 310 K and 1.01325 bar using Desmond software[52].

**Statistics and reproducibility**. Data were analyzed by one-way analysis of variance followed by Dunnet's multiple-comparison post hoc test using Prism 8 (GraphPad Software, CA, USA). Statistical tests and sample sizes used in the experiments are described in the corresponding figure and table legends. Experiments were replicated to confirm reproducibility.

**Reporting summary**. Further information on research design is available in the Nature Research Reporting Summary linked to this article.

## Data availability

All the data supporting findings of the research are available in Figs. 1–5, Supplementary Figs. 1–18 and Supplementary Tables 1–3. Source data are available in Supplementary Data 1. Additional supporting data and detailed expression methods including full plasmid sequences are able to be provided by the corresponding author upon reasonable requests.

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

## Acknowledgements

The authors thank Fumie Futaki, Michiya Kanzaki, and Shigeru Kawahara, Ajinomoto Co., Inc. for contribution to commenting on the study. The authors also thank Keiko Mizukami, Toshihiro Toba, and Mayuko Abe, WDB Co. Ltd., for their kind support in performing experiments. The authors would like to thank Enago (www.enago.jp) for the English language review.

## Author contributions

The study was conceptualized by K.I., Y.K., and A.K. All the experiments were planned and conducted by K.I. K.I. designed peptides and carried cell-based assays. K.I. and K.M. carried SPR analysis. Y.M. and Y.K. carried bacterial expressions. A.M. carried LC–MS analysis and purification of the peptides. N.S. and K.S. performed disulfide bond analysis. K.T. carried MD simulation. The paper was written by K.I. and modified on and discussed by all authors.

## Competing interests

K.I., Y.M., A.M., K.M., K.T., Y.K., and A.K. declare a potential conflict of interest with the patent WO2021/112249. The other authors declare no competing interests.
