## [Peer Review File · Communications Biology]

Reviewers' comments:

Reviewer #1 (Remarks to the Author):

The manuscript by Ito and coworkers describes a novel format of growth factor/cytokine mimetics. The authors propose that peptides composed of two or more tandemly linked macrocyclic peptides binding to growth factor/cytokine receptors are useful growth factor/cytokine mimetics. The peptides, named STaMPtides, can be expressed in and secreted from bacterial cells, thus STaMPtides can be used for assays without laborious purification steps. They have demonstrated the bioactivity of STaMPtides for three cases, i.e. HGF, EPO and TPO mimetics. The authors present a wide range of data ranging from peptide characterization and cellular assay results. Given the importance of growth factor mimetics and the wide applicability of the STaMPtide methodology, the manuscript should be considered for publication in Communications Biology. However, before acceptance of the manuscript, the authors should address the following points.

- 1) The authors should assess the effect of HGF-mimetic aMD4dY-PA22 on the phosphorylation level of MET. In the submitted manuscript, the effect of aMD4dY-PA22 on MET is indirectly assessed by a serum response element assay monitoring Erk signaling activation. However, considering the complexity of cellular phosphorylation pathway, the authors should assess the effect of the HGF-mimetic more directly on MET phosphorylation level by ELISA and western blotting in comparison with HGF.
- 2) The evaluation of the effect of aMD4dY-PA22 on BM-hMSC culture is too primitive. Only morphological change is chased in the manuscript, but expression of marker proteins, e.g. CD45, CD90, CD105, and secretion of cytokines that are characteristic to BM-hMSCs and differentiated cells should be evaluated.

Reviewer #2 (Remarks to the Author):

Ito and colleagues describe an elegant strategy to design and produce bi- and trivalent peptide ligands as ligands for growth factor receptors. Such multivalent peptide macrocycles have been proven to be valuable mimetics of cytokines/growth factors. Three examples are presented with new ligands for human Met Tyr Kinase (hepatocyte growth factor receptor), EPO receptor and TPO receptor. Using known macrocyclic ligands as a starting point the authors optimized them for heterologous production and verified the anticipated structure and activity of the expressed ligand in several assays. Overall this is a well written manuscript that presents an interesting novel strategy to develop multivalent macrocyclic peptide as research tools and/or therapeutics. I only have a few minor comments that the authors should consider:

- 1) The author use *Corynebacterium glutamicum* as an expression host for their studies. Can this strategy be adapted to more common expression host (*E.coli*, *B. subtilis*, *Pichia pastoris*) ? Perhaps this could be discussed in the Discussion.
- 2) An important part of the development process in this paper is the choice of linker. The authors settled on poly(Pro-Ala). Is this a validated linker that is sufficiently biocompatible ? I am thinking about stability in vivo, immunogenicity etc. ?
- 3) Maybe I missed it, but would the authors be able to comment on the yield of the expressed peptides (mg/mL of culture) ideally after purification ?
- 4) In Figure S6, please indicate the number of biological replicates? I understand this is a rapid, preliminary screen so I assume n=1? This is fine, just indicate in Figure legend. In Figure S6b, why does the reported activity peak at 10^2 – 10^3 dilution ? Or why is activity reduced at very low dilutions (high concentrations)?

Reviewer #3 (Remarks to the Author):

The paper describes an interesting method for producing a tandem repeat of a small disulfide-containing peptide. The production method itself is not novel nor is the peptide. Instead the central thesis of this work is that tandem repeat conjugation of the two looped peptides (macrocycles) results in improved activity compared to the monomer. Unfortunately the data provided does not support this assertion.

The improvement in activity is modest. The gain observed is between 10^{-1} and 10^{-2} for a ligand that is as a monomer 10^{-8} or 10^{-9} . In theory, optimal orientation of a dimer would result in dissociation constants closer to 10^{-16} to 10^{-18} . In the case of aMD4dY it is argued that the dissociation is slow due to dimerisation whereas in EMP35 this is in fact worse and instead it is argued that pre-organisation is the theoretical explanation for the improved on-rate. There is no evidence provided in support of either claim (and without replicates or errors, the significance of these difference in the SPR data is impossible to assess).

The argument regarding pre-organisation of a dimer is particularly difficult to justify in the current design. The number of rotatable bonds, hence degrees-of-freedom and available entropy, would effectively reject this as a possibility on a theoretical basis.

In my view the only way to support the claim of the paper is to perform the binding assays where the monomer and dimer are compared in a more suitable way. I can suggest the following:

AET - CRQFNRRTHEVWNLDCG - AAPAAPA PAAPAAPA PAAPAA PG - CRQFNRRTHEVWNLDC
compared to

AET - CRQFNRRTHEVWNLDCG - AAPAAPA PAAPAAPA PAAPAA PG

if gains due to the tandem repeat is being tested

or possibly to compare the above to

AET - CRQFNRRTHEVWNLDCG - AAPAAPA PAAPAAPA PAAPAA PG - SRQFNRRTHEVWNLDS

specifically if macrocyclisation is being tested in addition (apologies if any of the linker or peptide sequences are incorrectly transcribed, but these were not provided in any editable format). Other (and perhaps better) sequence are also possible (i.e. where the sequence of the second domain is jumbled to generate a peptide that would have the same chemical properties but a different receptor binding interface - or if the pharmacophore is known perhaps a binding incompetent second macrocycle).

The same would be required for the EMP35.

The reason I'm hesitant to accept the justification by authors is that the modest gains observed may simply be due to some favourable interactions between some components of the linker and the receptor (or indeed the SPR chip or the cell membrane). If these interactions are in the order of 10^{-2} in affinity (mM dissociation constants), then this may be sufficient to explain their observations.

The remainder of the work is done using crudely purified materials and I hesitate to make any quantitative conclusions based on these assays (which I'm not personally familiar with).

Some minor issues to also consider in future work.

What is the pH of the enzyme digestion? At high pH there will be disulfide shuffling in aqueous solution. What is the source of the enzymes.

What were the results from the GS linked molecules. Were these degraded prior to purification?

It would be helpful to compare these molecules with their parent compounds using the same assay, it is stated in the original publication by Suga that 10^{-2} reduction in activity is observed when the macrocycle is made linear. This seems similar to the difference being observed here. In general comparison to the original monomer and dimer would significantly improve this manuscript as it would allow assessment of the viability of the method to replicate results that are achieved by other approaches to dimerisation of macrocycles.

Point-to-point response to reviews

We thank the reviewers for their helpful comments. We have responded to the comments point by point, adding results, discussion, and references as follows. We hope this has made the manuscript clearer and easier to understand for the readers.

Reviewer #1:

The manuscript by Ito and coworkers describes a novel format of growth factor/cytokine mimetics. The authors propose that peptides composed of two or more tandemly linked macrocyclic peptides binding to growth factor/cytokine receptors are useful growth factor/cytokine mimetics. The peptides, named STaMPtides, can be expressed in and secreted from bacterial cells, thus STaMPtides can be used for assays without laborious purification steps. They have demonstrated the bioactivity of STaMPtides for three cases, i.e. HGF, EPO and TPO mimetics. The authors present a wide range of data ranging from peptide characterization and cellular assay results. Given the importance of growth factor mimetics and the wide applicability of the STaMPtide methodology, the manuscript should be considered for publication in Communications Biology.

However, before acceptance of the manuscript, the authors should address the following points.

We appreciated Reviewer 1's helpful comments and have added several cellular assays to more clearly demonstrate the applicability of STaMPtides in cell culture.

1) The authors should assess the effect of HGF-mimetic aMD4dY-PA22 on the phosphorylation level of MET. In the submitted manuscript, the effect of aMD4dY-PA22 on MET is indirectly assessed by a serum response element assay monitoring Erk signaling activation. However, considering the complexity of cellular phosphorylation pathway, the authors should assess the effect of the HGF-mimetic more directly on MET phosphorylation level by ELISA and western blotting in comparison with HGF.

As recommended, we performed previously reported phospho-MET ELISA (ref. 6, 32 in the manuscript) to quantify the phosphorylation level of MET in aMD4dY-PA22-stimulated HuCCT-1 cells in comparison with those in recombinant human HGF (rhHGF)-stimulated cells. aMD4dY-PA22 induced MET phosphorylation at

the same level as rhHGF (**newly added as Fig. 3a**), which is comparable to the reporter assay of Erk signaling activation.

We added the experimental method (page 23, line 495-page 24 line 511) and result (page 8, line 163-page 9, line 173).

2) The evaluation of the effect of aMD4dY-PA22 on BM-hMSC culture is too primitive. Only morphological change is chased in the manuscript, but expression of marker proteins, e.g. CD45, CD90, CD105, and secretion of cytokines that are characteristic to BM-hMSCs and differentiated cells should be evaluated.

We analyzed the characteristics of BM-hMSCs cultured in the presence or absence of rhHGF or aMD4dY-PA22 through the two recommended evaluations as follows:

(i) We quantified the expressions of MSC marker proteins (CD73, CD90, and CD105 positive and CD34 and CD45 negative) via flow cytometry (**newly added as Fig. S12a, b**). The maintenance of the undifferentiated state of BM-hMSCs under the stimulation of rhHGF or aMD4dY-PA22 was confirmed by the consistent expression of MSC marker expression.

(ii) Cytokine secretome was analyzed by cytokine array. The conditioned media (supernatants) of BM-hMSCs with or without stimulants (rhHGF or aMD4dY-PA22) was analyzed by commercially available cytokine array, which can detect 80 cytokines. Cytokine secretion profiles were consistent regardless of the addition of rhHGF or aMD4dY-PA22 (**newly added as Fig. S12c, d**).

From these two evaluations, we conclude that aMD4dY-PA22 promoted the proliferation of BM-hMSCs without losing the characteristics of MSCs. We added sentences that describe these methods (page 24, line 528- page25, line 546), results and discussions (page 10, line 206-page11, line 219).

Reviewer #2:

Ito and colleagues describe an elegant strategy to design and produce bi- and trivalent peptide ligands as ligands for growth factor receptors. Such multivalent peptide macrocycles have been proven to be valuable mimetics of cytokines/growth factors. Three examples are presented with new ligands for human Met Tyr Kinase (hepatocyte growth factor receptor), EPO receptor and TPO receptor. Using known macrocyclic ligands as a starting point the authors optimized them for heterologous production and verified the anticipated structure and activity of the expressed ligand in several assays. Overall this is a well written manuscript that presents an interesting and novel strategy to develop multivalent macrocyclic peptide as research tools and/or therapeutics.

I only have a few minor comments that the authors should consider:

We appreciate the comments made by Reviewer 2. Accordingly, we added information to clarify our technology so that it is easier for readers to understand.

1) The author use *Corynebacterium glutamicum* as an expression host for their studies. Can this strategy be adapted to more common expression host (*E.coli*, *B. subtilis*, *Pichia pastoris*) ? Perhaps this could be discussed in the Discussion.

This STaMPtide strategy may be applicable to other expression systems. However, the *C. glutamicum*-based expression system may be more suitable than other expression systems owing to the following features:

(i) *C. glutamicum* efficiently secretes STaMPtides into the supernatant in an active form with appropriate folding and disulfide formation. In addition, it does not degrade short STaMPtides without expression tags/cargoes. Thus, no additional post-translational processes (macrocyclization, refolding, or tag/cargo-cleavage) are required.

(ii) *C. glutamicum* hardly secretes host cell proteins or endotoxin that would cause a nonspecific effect or toxicity in cellular assays.

We added the above information to the Discussion section (page 15, line 311-322).

2) An important part of the development process in this paper is the choice of linker. The authors settled on poly(Pro-Ala). Is this a validated linker that is sufficiently biocompatible ? I am thinking about stability in vivo, immunogenicity etc. ?

We have not tested the stability of poly(Pro-Ala) in animals but assumed them to be sufficiently stable *in vivo* because poly(Pro-Ala) was originally developed to stabilize

proteins/peptides *in vivo*. Also, poly(Pro-Ala) is reported to be slightly immunogenic in a previous report. These features have been indicated with a reference (ref. 22) in the original manuscript (page 6, line 98-100).

We also added another reference (ref. 23) that clearly describes the above features of poly(Pro-Ala).

3) Maybe I missed it, but would the authors be able to comment on the yield of the expressed peptides (mg/mL of culture) ideally after purification ?

We added information about the expression yield of STaMPtides (**newly added as Supplementary Table S3**). Each of the expression yields was calculated by quantifying the peptide amount in the culture broth using LC-MS. We added an experimental method in the Materials and Methods section (page 20, line 435-437).

4) In Figure S6, please indicate the number of biological replicates? I understand this is a rapid, preliminary screen so I assume n=1? This is fine, just indicate in Figure legend. In Figure S6b, why does the reported activity peak at 10² – 10³ dilution ? Or why is activity reduced at very low dilutions (high concentrations)?

The biological replicates of Figure S6 were n = 1. We have added this information to the figure legend of Supplementary Figure S6.

Met is activated when it forms homodimers. The activity of aMD4dY-PA22 is reduced at a high concentration because aMD4dY-PA22 and Met form a 1:1 inactive complex with an excess amount of aMD4dY-PA22, whereas they form a 1:2 active complex with an appropriate amount. Because this bell-shaped activity is well-discussed using calculation models in previous reports, we added a simple description with a citation (page 7, line 137-139).

Reviewer #3:

The paper describes an interesting method for producing a tandem repeat of a small disulfide-containing peptide. The production method itself is not novel nor is the peptide. Instead the central thesis of this work is that tandem repeat conjugation of the two looped peptides (macrocycles) results in improved activity compared to the monomer. Unfortunately the data provided does not support this assertion.

We appreciate the comments given by Reviewer 3. First, we apologize that the central thesis was unclear in the original manuscript, but it is not an improvement in affinity but a new strategy for activating the receptor tyrosine kinases using a new bivalent peptide format.

To avoid emphasizing an improvement of affinity and to clarify the central thesis of this work clearer, we moved the table of kinetic parameters (**Table 1** in the original manuscript) to **Supplementary Table 2**.

Furthermore, in response to concerns about the manuscript, we are responding with the following point-by-point comments.

The improvement in activity is modest. The gain observed is between 10^{-1} and 10^{-2} for a ligand that is as a monomer 10^{-8} or 10^{-9} . In theory, optimal orientation of a dimer would result in dissociation constants closer to 10^{-16} to 10^{-18} . In the case of aMD4dY it is argued that the dissociation is slow due to dimerisation whereas in EMP35 this is in fact worse and instead it is argued that pre-organisation is the theoretical explanation for the improved on-rate. There is no evidence provided in support of either claim (and without replicates or errors, the significance of these difference in the SPR data is impossible to assess).

The argument regarding pre-organisation of a dimer is particularly difficult to justify in the current design. The number of rotatable bonds, hence degrees-of-freedom and available entropy, would effectively reject this as a possibility on a theoretical basis.

First, we interpreted the “increase in activity” mentioned in the comment above as an “increase in kinetic properties.” As the binding model of bivalent and multivalent ligands is different from that of monovalent ones, dimerization cannot improve the binding affinity by simple multiplication (please see the original and newly added references 26, 27). As previously reported (please see newly added ref 19, 28–31), the dimerization of ligands (peptides, VHH, and antibodies) improves the binding ability only modestly (improvement by $\sim 10^{-1}$ to 10^{-3}), which is consistent with our results. We added this information and the above references in the manuscript (page8, line

155-160).

It is difficult to experimentally justify the assumption about the pre-organization of macrocyclic moieties of EMP-STaMPtides by SPR because the improvement of the kinetic parameter via dimerization is relatively small (~10-fold-improvement), although peptides have strong binding potencies, which are almost at the measurement limit. Furthermore, this discussion is not the main focus of this research. Thus, we deleted the discussion about pre-organization and re-constructed the paragraph to discuss the binding potency and activity of EPO-mimetic STaMPtides (page 12, line 244-257).

In my view the only way to support the claim of the paper is to perform the binding assays where the monomer and dimer are compared in a more suitable way. I can suggest the following:

AET - CRQFNRRTHEVWNLDCG - AAPAAPA PAAPAAPA PAAPAA PG -
CRQFNRRTHEVWNLDC

compared to

AET - CRQFNRRTHEVWNLDCG - AAPAAPA PAAPAAPA PAAPAA PG

if gains due to the tandem repeat is being tested

or possibly to compare the above to

AET - CRQFNRRTHEVWNLDCG - AAPAAPA PAAPAAPA PAAPAA PG -
SRQFNRRTHEVWNLDS

specifically if macrocyclisation is being tested in addition (apologies if any of the linker or peptide sequences are incorrectly transcribed, but these were not provided in any editable format). Other (and perhaps better) sequence are also possible (i.e. where the sequence of the second domain is jumbled to generate a peptide that would have the same chemical properties but a different receptor binding interface - or if the pharmacophore is known perhaps a binding incompetent second macrocycle).

The same would be required for the EMP35.

The reason I'm hesitant to accept the justification by authors is that the modest gains observed may simply be due to some favourable interactions between some components of the linker and the receptor (or indeed the SPR chip or the cell membrane). If these interactions are in the order of 10⁻² in affinity (mM dissociation constants), then this may be sufficient to explain their observations.

Thank you for the comments and suggestions for further experiments. We understand

that the reviewer has the following two concerns in the improvement of kinetic properties: (i) the contribution of linkers and (ii) contribution of non-macrocyclized (or nonspecific) weak binders to the affinity.

Regarding the (i) contribution of linkers, we propose that linkers rarely contribute to an improvement in affinity. As given in our response to the above comments, the dimerization of ligands with 10^{-9} affinity improves to only 10^{-1} to 10^{-3} affinity. Thus, a negligible improvement in affinity should occur with only linker conjugation, even if it would have nonspecific weak interaction. Furthermore, we introduced several reports that demonstrate that the conjugation of the linker to a peptide does not alter affinity (please see ref. 36 in the manuscript and Noberini R, *et al.*, *PLoS ONE*. 6, e28611, 2011.).

Regarding (ii) the contribution of non-macrocyclized (or nonspecific) weak binding sites to the affinity, STaMPtides contains two macrocyclized sites, which was confirmed by proteolytic disulfide analysis. As there are no nonspecific binding sites, we do not need to consider weak interactions but only receptor-specific interactions by macrocyclic sites.

The remainder of the work is done using crudely purified materials and I hesitate to make any quantitative conclusions based on these assays (which I'm not personally familiar with).

Thank you for your comments concerning quantitative analysis; however, crude materials were utilized for screening assays, whereas purified materials were used in the main assays for the quantitative evaluation of STaMPtides.

This is clearly stated in the original manuscript (aMD4dY-PA22: page 8, line 146-148. EMP-PA8/22: page 12, line 241-243. TMP-PA8: page 14, line 286-289).

Some minor issues to also consider in future work.

What is the pH of the enzyme digestion? At high pH there will be disulfide shuffling in aqueous solution. What is the source of the enzymes.

Thank you for your critical comments. We were concerned about disulfide shuffling, so we digested the samples under acidic condition. The pH was 6.6 for thermolysin digestion and 1.5 for ProAlanase digestion. Enzymes were isolated and purified from *Geobacillus stearothermophilus* (thermolysin) or fungus *Aspergillus niger* (ProAlanase); both are commercially available at Promega as written in the "Materials" section. To clarify the experimental procedure, information about pH was added to the Experimental section (page 21, line 441-443).

What were the results from the GS linked molecules. Were these degraded prior to purification?

Thank you for your advice on this point. We added results of the activity assay using aMD4dY dimers with GS linkers as **Supplementary Fig. S6(c)** to the Results section. As they showed agonistic activities, they were not fully degraded in the *C. glutamicum* culture broth.

It would be helpful to compare these molecules with their parent compounds using the same assay, it is stated in the original publication by Suga that 10⁻² reduction in activity is observed when the macrocycle is made linear. This seems similar to the difference being observed here. In general comparison to the original monomer and dimer would significantly improve this manuscript as it would allow assessment of the viability of the method to replicate results that are achieved by other approaches to dimerisation of macrocycles.

We will address the Reviewer's two concerns as follows:

- (i) Contributions of dimerization and macrocyclization of peptides to their activity (affinity)
- (ii) Applicability of the strategy of converting macrocycle dimers developed by other dimerization methods with STaMPtides

Regarding concern (i), we first emphasize that the mechanisms of contributions to the affinity of dimerization and macrocyclization differ. Dimerization improves affinity via the avidity effect of the bivalent ligand, as was commented above, and macrocyclization via the lower free energy of organizing an active structure (please see newly added ref. 14). As pursuing the mechanism of affinity improvement by bivalency and macrocyclization are beyond the main scope of the manuscript, as described above, we would like to skip performing the proposed experiment.

For (ii), we demonstrated three examples (HGF-mimetic, EPO-mimetic, and TPO-mimetic dimers) in the manuscript and believe that the viability and versatility of the approach has been well demonstrated. However, to address the concern about the applicability of our approach, we provided another example of STaMPtide-based dimerization of macrocycles. In a previous report (A. Shrivastava, *et al.*, *Protein Eng. Des. Sel.* 18, 9, 417–24, 2015), two respective macrocyclic peptides that bind to VEGF receptor (VEGFR) were identified and dimerized using a chemical linker. We applied

the STaMPtide format to this peptide dimer and confirmed the improved binding potency and inhibitory activity owing to the VEGF–VEGFR interaction. Because this result consists of unpublished data and is beyond the main focus of this manuscript, we have attached it as a **Figure for Review** below. Considering the three-growth factor and cytokine mimetics and one inhibitory peptide together, we believe that the STaMPtide strategy is well justified.

FIGURE REDACTED

Figure for Review REDACTED

Reviewers' comments:

Reviewer #1 (Remarks to the Author):

The authors provided sufficient additional data and explanation in response to the reviewers' concerns. Therefore, the manuscript is recommended for a publication in Communications Biology. Additional comment: On the newly added graph in Fig. 3a, the phosphorylation level of Met upon stimulation by aMD4dY-PA22 seems going to exceed the level upon stimulation by the native ligand rhHGF. Measurement of the Met phosphorylation level upon stimulation by higher concentrations of aMD4dY-PA22 may be an interesting future experiment for the authors.

Reviewer #2 (Remarks to the Author):

The authors have argued well and have addressed all of my concerns. The present version of the manuscript is a significant improvement over the initially submitted manuscript and, in my opinion, is suitable as a research article for Communications Biology.

Reviewer #3 (Remarks to the Author):

The authors have made several improvements on their manuscript. To avoid further confusion and prolonged discussions I have below outlined two critical points that require attention prior to publication.

1. The authors reject the possibility of linker interactions without providing any evidence of this. This is speculation and should be stated clearly as such in the manuscript. I.e. there must be a statement along the lines of "although we consider this unlikely, we cannot rule out that the improvements we are seeing are due to the linker sequence rather than the second macrocycle." I have already stated what experiments would be required to rule out the linker as a potential source of their observations. One very curious omission that would also resolve this issue is to measure kinetic parameters for the GS linked dimer.
2. The kinetic parameters in Table S1 have no replicated and therefore no estimates of the experimental uncertainty. This is not acceptable.

Point-to-point response to reviewer

We thank the reviewer for this helpful comment. Please find below our response to the comment and corresponding revisions in the manuscript.

Reviewer #3 (Remarks to the Author):

2. The kinetic parameters in Table S1 have no replicated and therefore no estimates of the experimental uncertainty. This is not acceptable.

We appreciate this suggestion. Accordingly, we measured kinetic parameters with $N = 3$. The newly calculated parameters from the triplicated experiments are given in **Supplementary Table S1** in terms of mean with standard deviation. We have made the following revisions in the text.

(Page 8)

The dissociation constant (K_D) of aMD4dY-PA22 with human Met was 1.0 nM (Supplementary Table S2), which was 45 times lower than that of monomeric aMD4dY and was consistent with previously reported bivalent molecules.

(Page 12)

All the constructs showed strong binding potency against EPOR. Although exact dissociation constants could not be determined because of their too slow dissociation, the association rate of EMP-PA8 and EMP-PA22 was revealed to be 5–10 times higher than that of EMP35 (Supplementary Table S2).